# Shape and Time Distortion Loss for Training Deep Time Series Forecasting Models

**Vincent Le Guen** [1,2]
vincent.le-guen@edf.fr

**Nicolas Thome** [2]
nicolas.thome@cnam.fr

(1) EDF R&D
6 quai Watier, 78401 Chatou, France

(2) CEDRIC, Conservatoire National des Arts et Métiers
292 rue Saint-Martin, 75003 Paris, France

## Abstract

This paper addresses the problem of time series forecasting for non-stationary signals and multiple future steps prediction. To handle this challenging task, we introduce DILATE (DIstortion Loss including shApe and TimE), a new objective function for training deep neural networks. DILATE aims at accurately predicting sudden changes, and explicitly incorporates two terms supporting precise shape and temporal change detection. We introduce a differentiable loss function suitable for training deep neural nets, and provide a custom back-prop implementation for speeding up optimization. We also introduce a variant of DILATE, which provides a smooth generalization of temporally-constrained Dynamic Time Warping (DTW). Experiments carried out on various non-stationary datasets reveal the very good behaviour of DILATE compared to models trained with the standard Mean Squared Error (MSE) loss function, and also to DTW and variants. DILATE is also agnostic to the choice of the model, and we highlight its benefit for training fully connected networks as well as specialized recurrent architectures, showing its capacity to improve over state-of-the-art trajectory forecasting approaches.

## 1  Introduction

Time series forecasting [6] consists in analyzing the dynamics and correlations between historical data for predicting future behavior. In one-step prediction problems [39, 30], future prediction reduces to a single scalar value. This is in sharp contrast with multi-step time series prediction [49, 2, 48], which consists in predicting a complete trajectory of future data at a rather long temporal extent. Multi-step forecasting thus requires to accurately describe time series evolution.

This work focuses on multi-step forecasting problems for non-stationary signals, *i.e.* when future data cannot only be inferred from the past periodicity, and when abrupt changes of regime can occur. This includes important and diverse application fields, *e.g.* regulating electricity consumption [63, 36], predicting sharp discontinuities in renewable energy production [23] or in traffic flow [35, 34], electrocardiogram (ECG) analysis [9], stock markets prediction [14], *etc*.

Deep learning is an appealing solution for this multi-step and non-stationary prediction problem, due to the ability of deep neural networks to model complex nonlinear time dependencies. Many approaches have recently been proposed, mostly relying on the design of specific one-step ahead

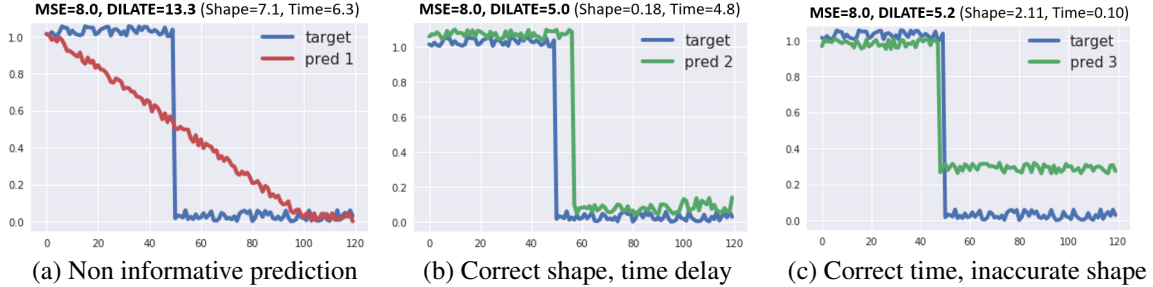

| (a) Non informative prediction | (b) Correct shape, time delay | (c) Correct time, inaccurate shape |

Figure 1: Limitation of the euclidean (MSE) loss: when predicting a sudden change (target blue step function), the 3 predictions (a), (b) and (c) have similar MSE but very different forecasting skills. In contrast, the DILATE loss proposed in this work, which disentangles shape and temporal decay terms, supports predictions (b) and (c) over prediction (a) that does not capture the sharp change of regime.

architectures recursively applied for multi-step [24, 26, 7, 5], on direct multi-step models [3] such as Sequence To Sequence [34, 60, 57, 61] or State Space Models for probabilistic forecasts [44, 40].

Regarding training, the huge majority of methods use the Mean Squared Error (MSE) or its variants (MAE, *etc*) as loss functions. However, relying on MSE may arguably be inadequate in our context, as illustrated in Fig 1. Here, the target ground truth prediction is a step function (in blue), and we present three predictions, shown in Fig 1(a), (b), and (c), which have a similar MSE loss compared to the target, but very different forecasting skills. Prediction (a) is not adequate for regulation purposes since it doesn't capture the sharp drop to come. Predictions (b) and (c) much better reflect the change of regime since the sharp drop is indeed anticipated, although with a slight delay (b) or with a slight inaccurate amplitude (c).

This paper introduces DILATE (DIstortion Loss including shApe and TimE), a new objective function for training deep neural networks in the context of multi-step and non-stationary time series forecasting. DILATE explicitly disentangles into two terms the penalization related to the shape and the temporal localization errors of change detection (section 3). The behaviour of DILATE is shown in Fig 1: whereas the values of our proposed shape and temporal losses are large in Fig 1(a), the shape (resp. temporal) term is small in Fig 1(b) (resp. Fig 1(c)). DILATE combines shape and temporal terms, and is consequently able to output a much smaller loss for predictions (b) and (c) than for (a), as expected.

To train deep neural nets with DILATE, we derive a differentiable loss function for both shape and temporal terms (section 3.1), and an efficient and custom back-prop implementation for speeding up optimization (section 3.2). We also introduce a variant of DILATE, which provides a smooth generalization of temporally-constrained Dynamic Time Warping (DTW) metrics [43, 28]. Experiments carried out on several synthetic and real non-stationary datasets reveal that models trained with DILATE significantly outperform models trained with the MSE loss function when evaluated with shape and temporal distortion metrics, while DILATE maintains very good performance when evaluated with MSE. Finally, we show that DILATE can be used with various network architectures and can outperform on shape and time metrics state-of-the-art models specifically designed for multi-step and non-stationary forecasting.

## 2 Related work

**Time series forecasting**  Traditional methods for time series forecasting include linear auto-regressive models, such as the ARIMA model [6], and Exponential Smoothing [27], which both fall into the broad category of linear State Space Models (SSMs) [17]. These methods handle linear dynamics and stationary time series (or made stationary by temporal differences). However the stationarity assumption is not satisfied for many real world time series that can present abrupt changes of distribution. Since, Recurrent Neural Networks (RNNs) and variants such as Long Short Term Memory Networks (LSTMs) [25] have become popular due to their automatic feature extraction abilities, complex patterns and long term dependencies modeling. In the era of deep learning, much effort has been recently devoted to tackle multivariate time series forecasting with a huge number of input

series [31], by leveraging attention mechanisms [30, 39, 50, 12] or tensor factorizations [60, 58, 46] for capturing shared information between series. Another current trend is to combine deep learning and State Space Models for modeling uncertainty [45, 44, 40, 56]. In this paper we focus on deterministic multi-step forecasting. To this end, the most common approach is to apply recursively a one-step ahead trained model. Although mono-step learned models can be adapted and improved for the multi-step setting [55], a thorough comparison of the different multi-step strategies [48] has recommended the direct multi-horizon strategy. Of particular interest in this category are Sequence To Sequence (Seq2Seq) RNNs models [1] [44, 31, 60, 57, 19] which achieved great success in machine translation. Theoretical generalization bounds for Seq2Seq forecasting were derived with an additional discrepancy term quantifying the non-stationarity of time series [29]. Following the success of WaveNet for audio generation [53], Convolutional Neural Networks with dilation have become a popular alternative for time series forecasting [5]. The self-attention Transformer architecture [54] was also lately investigated for accessing long-range context regardless of distance [32]. We highlight that our proposed loss function can be used for training any direct multi-step deep architecture.

**Evaluation and training metrics**   The largely dominant loss function to train and evaluate deep models is the MAE, MSE and its variants (SMAPE, etc). Metrics reflecting shape and temporal localization exist: Dynamic Time Warping [43] for shape ; timing errors can be casted as a detection problem by computing Precision and Recall scores after segmenting series by Change Point Detection [8, 33], or by computing the Hausdorff distance between two sets of change points [22, 51]. For assessing the detection of ramps in wind and solar energy forecasting, specific algorithms were designed: for shape, the ramp score [18, 52] based on a piecewise linear approximation of the derivatives of time series; for temporal error estimation, the Temporal Distortion Index (TDI) [20, 52]. However, these evaluation metrics are not differentiable, making them unusable as loss functions for training deep neural networks. The impossibility to directly optimize the appropriate (often non-differentiable) evaluation metric for a given task has bolstered efforts to design good surrogate losses in various domains, for example in ranking [15, 62] or computer vision [38, 59].

Recently, some attempts have been made to train deep neural networks based on alternatives to MSE, especially based on a smooth approximation of the Dynamic time warping (DTW) [13, 37, 1]. Training DNNs with a DTW loss enables to focus on the shape error between two signals. However, since DTW is by design invariant to elastic distortions, it completely ignores the temporal localization of the change. In our context of sharp change detection, both shape and temporal distortions are crucial to provide an adequate forecast. A differentiable timing error loss function based on DTW on the event (binary) space was proposed in [41] ; however it is only applicable for predicting binary time series. This paper specifically focuses on designing a loss function able to disentangle shape and temporal delay terms for training deep neural networks on real world time series.

## 3   Training Deep Neural Networks with DILATE

Our proposed framework for multi-step forecasting is depicted in Figure 2. During training, we consider a set of $N$ input time series $\mathcal{A} = \{\mathbf{x}_i\}_{i \in \{1:N\}}$. For each input example of length $n$, i.e. $\mathbf{x}_i = (\mathbf{x}_i^1, ..., \mathbf{x}_i^n) \in \mathbb{R}^{p \times n}$, a forecasting model such as a neural network predicts the future $k$-step ahead trajectory $\hat{\mathbf{y}}_i = (\hat{\mathbf{y}}_i^1, ..., \hat{\mathbf{y}}_i^k) \in \mathbb{R}^{d \times k}$. Our DILATE objective function, which compares this prediction $\hat{\mathbf{y}}_i$ with the actual ground truth future trajectory $\overset{*}{\mathbf{y}}_i = (\overset{*}{\mathbf{y}}_i^1, ..., \overset{*}{\mathbf{y}}_i^k)$ of length $k$, is composed of two terms balanced by the hyperparameter $\alpha \in [0, 1]$:

$$\mathcal{L}_{DILATE}(\hat{\mathbf{y}}_i, \overset{*}{\mathbf{y}}_i) = \alpha \, \mathcal{L}_{shape}(\hat{\mathbf{y}}_i, \overset{*}{\mathbf{y}}_i) + (1 - \alpha) \, \mathcal{L}_{temporal}(\hat{\mathbf{y}}_i, \overset{*}{\mathbf{y}}_i) \qquad (1)$$

**Notations and definitions**   Both our shape $\mathcal{L}_{shape}(\hat{\mathbf{y}}_i, \overset{*}{\mathbf{y}}_i)$ and temporal $\mathcal{L}_{temporal}(\hat{\mathbf{y}}_i, \overset{*}{\mathbf{y}}_i)$ distortions terms are based on the alignment between predicted $\hat{\mathbf{y}}_i \in \mathbb{R}^{d \times k}$ and ground truth $\overset{*}{\mathbf{y}}_i \in \mathbb{R}^{d \times k}$ time series. We define a warping path as a binary matrix $\mathbf{A} \subset \{0, 1\}^{k \times k}$ with $A_{h,j} = 1$ if $\hat{\mathbf{y}}_i^h$ is associated to $\overset{*}{\mathbf{y}}_i^j$, and 0 otherwise. The set of all valid warping paths connecting the endpoints $(1, 1)$ to $(k, k)$

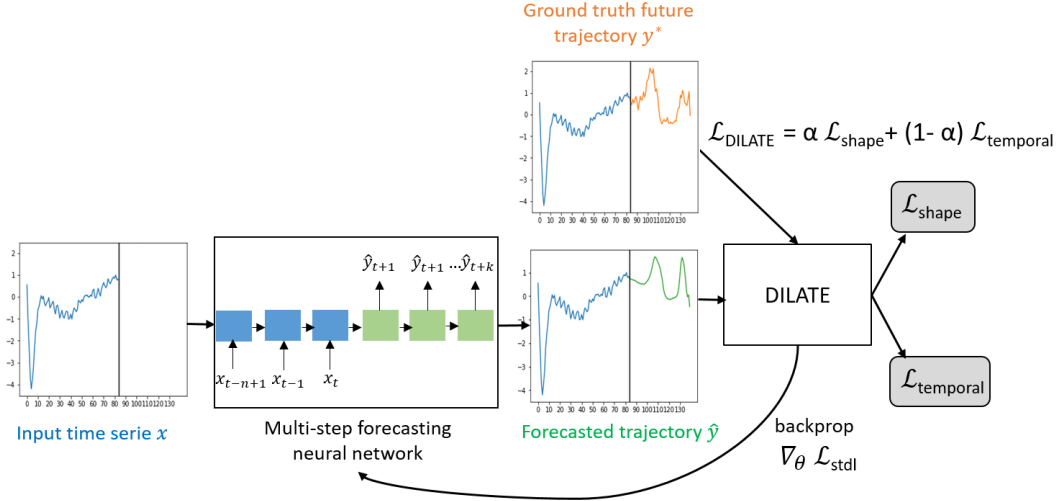

Figure 2: Our proposed framework for training deep forecasting models.

with the authorized moves $\rightarrow, \downarrow, \searrow$ (step condition) is noted $\mathcal{A}_{k,k}$. Let $\boldsymbol{\Delta}(\hat{\mathbf{y}}_i, \overset{*}{\mathbf{y}}_i) := [\delta(\hat{\mathbf{y}}_i^h, \overset{*}{\mathbf{y}}_i^j)]_{h,j}$ be the pairwise cost matrix, where $\delta$ is a given dissimilarity between $\hat{\mathbf{y}}_i^h$ and $\overset{*}{\mathbf{y}}_i^j$, *e.g.* the euclidean distance.

## 3.1 Shape and temporal terms

**Shape term**    Our shape loss function is based on the Dynamic Time Warping (DTW) [43], which corresponds to the following optimization problem: $DTW(\hat{\mathbf{y}}_i, \overset{*}{\mathbf{y}}_i) = \min_{\mathbf{A} \in \mathcal{A}_{k,k}} \left\langle \mathbf{A}, \boldsymbol{\Delta}(\hat{\mathbf{y}}_i, \overset{*}{\mathbf{y}}_i) \right\rangle$.

$\mathbf{A}^* = \underset{\mathbf{A} \in \mathcal{A}_{k,k}}{\arg\min} \left\langle \mathbf{A}, \boldsymbol{\Delta}(\hat{\mathbf{y}}_i, \overset{*}{\mathbf{y}}_i) \right\rangle$ is the optimal association (path) between $\hat{\mathbf{y}}_i$ and $\overset{*}{\mathbf{y}}_i$. By temporally

aligning the predicted $\hat{\mathbf{y}}_i$ and ground truth $\overset{*}{\mathbf{y}}_i$ time series, the DTW loss focuses on the structural shape dissimilarity between signals. The DTW, however, is known to be non-differentiable. We use the smooth min operator $\min_\gamma(a_1, ..., a_n) = -\gamma \log(\sum_i^n \exp(-a_i/\gamma))$ with $\gamma > 0$ proposed in [13] to define our differentiable shape term $\mathcal{L}_{shape}$:

$$\mathcal{L}_{shape}(\hat{\mathbf{y}}_i, \overset{*}{\mathbf{y}}_i) = DTW_\gamma(\hat{\mathbf{y}}_i, \overset{*}{\mathbf{y}}_i) := -\gamma \log \left( \sum_{\mathbf{A} \in \mathcal{A}_{k,k}} \exp \left( -\frac{\left\langle \mathbf{A}, \boldsymbol{\Delta}(\hat{\mathbf{y}}_i, \overset{*}{\mathbf{y}}_i) \right\rangle}{\gamma} \right) \right) \quad (2)$$

**Temporal term**    Our second term $\mathcal{L}_{temporal}$ in Eq (1) aims at penalizing temporal distortions between $\hat{\mathbf{y}}_i$ and $\overset{*}{\mathbf{y}}_i$. Our analysis is based on the optimal DTW path $\mathbf{A}^*$ between $\hat{\mathbf{y}}_i$ and $\overset{*}{\mathbf{y}}_i$. $\mathbf{A}^*$ is used to register both time series when computing DTW and provide a time-distortion invariant loss. Here, we analyze the form of $\mathbf{A}^*$ to compute the temporal distortions between $\hat{\mathbf{y}}_i$ and $\overset{*}{\mathbf{y}}_i$. More precisely, our loss function is inspired from computing the Time Distortion Index (TDI) for temporal misalignment estimation [20, 52], which basically consists in computing the deviation between the optimal DTW path $\mathbf{A}^*$ and the first diagonal. We first rewrite a generalized TDI loss function with our notations:

$$TDI(\hat{\mathbf{y}}_i, \overset{*}{\mathbf{y}}_i) = \langle \mathbf{A}^*, \boldsymbol{\Omega} \rangle = \left\langle \underset{\mathbf{A} \in \mathcal{A}_{k,k}}{\arg\min} \left\langle \mathbf{A}, \boldsymbol{\Delta}(\hat{\mathbf{y}}_i, \overset{*}{\mathbf{y}}_i) \right\rangle, \boldsymbol{\Omega} \right\rangle \quad (3)$$

where $\boldsymbol{\Omega}$ is a square matrix of size $k \times k$ penalizing each element $\hat{\mathbf{y}}_i^h$ being associated to an $\overset{*}{\mathbf{y}}_i^j$, for $h \neq j$. In our experiments we choose a squared penalization, *e.g.* $\boldsymbol{\Omega}(h, j) = \frac{1}{k^2}(h - j)^2$, but other

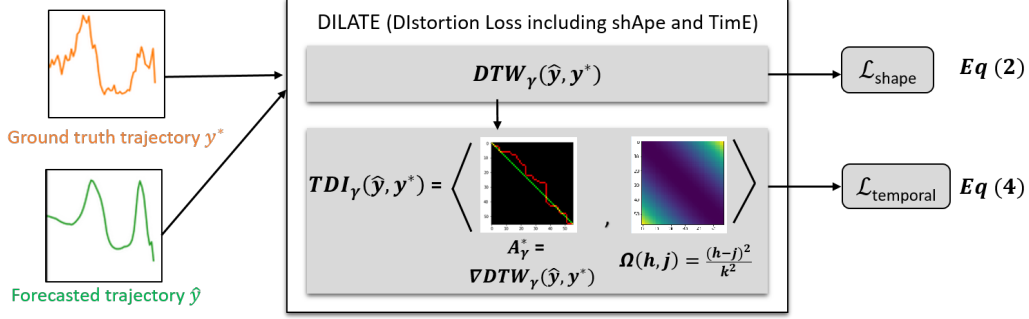

Figure 3: DILATE loss computation for separating the shape and temporal errors.

variants could be used. Note that *prior* knowledge can also be incorporated in the $\boldsymbol{\Omega}$ matrix structure, *e.g.* to penalize more heavily late than early predictions (and *vice versa*).

The TDI loss function in Eq (3) is still non-differentiable. Here, we cannot directly use the same smoothing technique that for defining $DTW_\gamma$ in Eq (2), since the minimization involves two different quantities $\boldsymbol{\Omega}$ and $\boldsymbol{\Delta}$. Since the optimal path $\mathbf{A}^*$ is itself non-differentiable, we use the fact that $\mathbf{A}^* = \nabla_\Delta DTW(\hat{\mathbf{y}}_i, \overset{*}{\mathbf{y}}_i)$ to define a smooth approximation $\mathbf{A}_\gamma^*$ of the $\arg\min$ operator, *i.e.* :

$$\mathbf{A}_\gamma^* = \nabla_\Delta DTW_\gamma(\hat{\mathbf{y}}_i, \overset{*}{\mathbf{y}}_i) = 1/Z \sum_{\mathbf{A}\in\mathcal{A}_{k,k}} \mathbf{A} \exp^{-\frac{\left\langle \mathbf{A}, \boldsymbol{\Delta}(\hat{\mathbf{y}}_i, \overset{*}{\mathbf{y}}_i) \right\rangle}{\gamma}}, \text{ with } Z = \sum_{\mathbf{A}\in\mathcal{A}_{k,k}} \exp^{-\frac{\left\langle \mathbf{A}, \boldsymbol{\Delta}(\hat{\mathbf{y}}_i, \overset{*}{\mathbf{y}}_i) \right\rangle}{\gamma}}$$

being the partition function. Based on $\mathbf{A}_\gamma^*$, we obtain our smoothed temporal loss from Eq (3):

$$\mathcal{L}_{temporal}(\hat{\mathbf{y}}_i, \overset{*}{\mathbf{y}}_i) := \left\langle \mathbf{A}_\gamma^*, \boldsymbol{\Omega} \right\rangle = \frac{1}{Z} \sum_{\mathbf{A}\in\mathcal{A}_{k,k}} \left\langle \mathbf{A}, \boldsymbol{\Omega} \right\rangle \exp^{-\frac{\left\langle \mathbf{A}, \boldsymbol{\Delta}(\hat{\mathbf{y}}_i, \overset{*}{\mathbf{y}}_i) \right\rangle}{\gamma}} \tag{4}$$

### 3.2 DILATE Efficient Forward and Backward Implementation

The direct computation of our shape and temporal losses in Eq (2) and Eq (4) is intractable, due to the cardinal of $\mathcal{A}_{k,k}$, which exponentially grows with $k$. We provide a careful implementation of the forward and backward passes in order to make learning efficient.

**Shape loss**   Regarding $\mathcal{L}_{shape}$, we rely on [13] to efficiently compute the forward pass with a variant of the Bellmann dynamic programming approach [4]. For the backward pass, we implement the recursion proposed in [13] in a custom Pytorch loss. This implementation is much more efficient than relying on vanilla auto-differentiation, since it reuses intermediate results from the forward pass.

**Temporal loss**   For $\mathcal{L}_{temporal}$, note that the bottleneck for the forward pass in Eq (4) is to compute $\mathbf{A}_\gamma^* = \nabla_\Delta DTW_\gamma(\hat{\mathbf{y}}_i, \overset{*}{\mathbf{y}}_i)$, which we implement as explained for the $\mathcal{L}_{shape}$ backward pass. Regarding $\mathcal{L}_{temporal}$ backward pass, we need to compute the Hessian $\nabla^2 DTW_\gamma(\hat{\mathbf{y}}_i, \overset{*}{\mathbf{y}}_i)$. We use the method proposed in [37], based on a dynamic programming implementation that we embed in a custom Pytorch loss. Again, our back-prop implementation allows a significant speed-up compared to standard auto-differentiation (see section 4.4).

The resulting time complexity of both shape and temporal losses for forward and backward is $\mathcal{O}(k^2)$.

**Discussion**   A variant of our approach to combine shape and temporal penalization would be to incorporate a temporal term inside our smooth $\mathcal{L}_{shape}$ function in Eq (2), *i.e.* :

$$\mathcal{L}_{DILATE^t}(\hat{\mathbf{y}}_i, \overset{*}{\mathbf{y}}_i) := -\gamma \log \left( \sum_{\mathbf{A}\in\mathcal{A}_{k,k}} \exp\left( -\frac{\left\langle \mathbf{A}, \alpha\boldsymbol{\Delta}(\hat{\mathbf{y}}_i, \overset{*}{\mathbf{y}}_i) + (1-\alpha)\boldsymbol{\Omega} \right\rangle}{\gamma} \right) \right) \tag{5}$$

We can notice that Eq (5) reduces to minimizing $\left\langle \mathbf{A}, \alpha\boldsymbol{\Delta}(\hat{\mathbf{y}}_i, \overset{*}{\mathbf{y}}_i) + (1-\alpha)\boldsymbol{\Omega} \right\rangle$ when $\gamma \to 0^+$. In this case, $\mathcal{L}_{DILATE^t}$ can recover DTW variants studied in the literature to bias the computation based on penalizing sequence misalignment, by designing specific $\boldsymbol{\Omega}$ matrices:

| | |
|---|---|
| Sakoe-Chiba DTW hard band constraint [43] | $\Omega(h,j) = +\infty$ if $|h-j| > T$, 0 otherwise |
| Weighted DTW [28] | $\Omega(h,j) = f(|i-j|)$, $f$ increasing function |

$\mathcal{L}_{DILATE^t}$ in Eq (5) enables to train deep neural networks with a smooth loss combining shape and temporal criteria. However, $\mathcal{L}_{DILATE^t}$ presents limited capacities for disentangling the shape and temporal errors, since the optimal path is computed from both shape and temporal terms. In contrast, our $\mathcal{L}_{DILATE}$ loss in Eq (1) separates the loss into two shape and temporal misalignment components, the temporal penalization being applied to the optimal unconstrained DTW path. We verify experimentally that our $\mathcal{L}_{DILATE}$ outperforms its "tangled" version $\mathcal{L}_{DILATE^t}$ (section 4.3).

## 4 Experiments

### 4.1 Experimental setup

**Datasets:** To illustrate the relevance of DILATE, we carry out experiments on 3 non-stationary time series datasets from different domains (see examples in Fig 4). The multi-step evaluation consists in forecasting the future trajectory on $k$ future time steps.

**Synthetic** ($k = 20$) dataset consists in predicting sudden changes (step functions) based on an input signal composed of two peaks. This controlled setup was designed to measure precisely the shape and time errors of predictions. We generate 500 times series for train, 500 for validation and 500 for test, with 40 time steps: the first 20 are the inputs, the last 20 are the targets to forecast. In each series, the input range is composed of 2 peaks of random temporal position $i_1$ and $i_2$ and random amplitude $j_1$ and $j_2$ between 0 and 1, and the target range is composed of a step of amplitude $j_2 - j_1$ and stochastic position $i_2 + (i_2 - i_1) + randint(-3, 3)$. All time series are corrupted by an additive gaussian white noise of variance 0.01.

**ECG5000** ($k = 56$) dataset comes from the UCR Time Series Classification Archive [10], and is composed of 5000 electrocardiograms (ECG) (500 for training, 4500 for testing) of length 140. We take the first 84 time steps (60 %) as input and predict the last 56 steps (40 %) of each time series (same setup as in [13]).

**Traffic** ($k = 24$) dataset corresponds to road occupancy rates (between 0 and 1) from the California Department of Transportation (48 months from 2015-2016) measured every 1h. We work on the first univariate series of length 17544 (with the same 60/20/20 train/valid/test split as in [30]), and we train models to predict the 24 future points given the past 168 points (past week).

**Network architectures and training:** We perform multi-step forecasting with two kinds of neural network architectures: a fully connected network (1 layer of 128 neurons), which does not make any assumption on data structure, and a more specialized Seq2Seq model [47] with Gated Recurrent Units (GRU) [11] with 1 layer of 128 units. Each model is trained with PyTorch for a max number of 1000 epochs with Early Stopping with the ADAM optimizer. The smoothing parameter $\gamma$ of DTW and TDI is set to $10^{-2}$. The hyperparameter $\alpha$ balancing $\mathcal{L}_{shape}$ and $\mathcal{L}_{temporal}$ is determined on a validation set to get comparable DTW shape performance than the $DTW_\gamma$ trained model: $\alpha = 0.5$ for Synthetic and ECG5000, and 0.8 for Traffic. Our code implementing DILATE is available on line from https://github.com/vincent-leguen/DILATE.

### 4.2 DILATE forecasting performances

We evaluate the performances of DILATE, and compare it against two strong baselines: the widely used Euclidean (MSE) loss, and the smooth DTW introduced in [13, 37]. For each experiment, we use the same neural network architecture (section 4.1), in order to isolate the impact of the training loss and to enable fair comparisons. The results are evaluated using three metrics: MSE, DTW (shape) and TDI (temporal). We perform a Student t-test with significance level 0.05 to highlight the best(s) method(s) in each experiment (averaged over 10 runs).

Overall results are presented in Table 1.

| Dataset | Eval | Fully connected network (MLP) | | | Recurrent neural network (Seq2Seq) | | |
|---------|------|-----|------|------|-----|------|------|
| | | MSE | DTW$_\gamma$ [13] | DILATE (ours) | MSE | DTW$_\gamma$ [13] | DILATE (ours) |
| Synth | MSE | **1.65 $\pm$ 0.14** | 4.82 $\pm$ 0.40 | **1.67$\pm$ 0.184** | **1.10 $\pm$ 0.17** | 2.31 $\pm$ 0.45 | **1.21 $\pm$ 0.13** |
| | DTW | 38.6 $\pm$ 1.28 | **27.3 $\pm$ 1.37** | 32.1 $\pm$ 5.33 | **24.6 $\pm$ 1.20** | 22.7 $\pm$ 3.55 | 23.1 $\pm$ 2.44 |
| | TDI | 15.3 $\pm$ 1.39 | 26.9 $\pm$ 4.16 | **13.8 $\pm$ 0.712** | 17.2 $\pm$ 1.22 | 20.0 $\pm$ 3.72 | **14.8 $\pm$ 1.29** |
| ECG | MSE | **31.5 $\pm$ 1.39** | 70.9 $\pm$ 37.2 | 37.2 $\pm$ 3.59 | **21.2 $\pm$ 2.24** | 75.1 $\pm$ 6.30 | 30.3 $\pm$ 4.10 |
| | DTW | 19.5 $\pm$ 0.159 | 18.4 $\pm$ 0.749 | **17.7 $\pm$ 0.427** | 17.8 $\pm$ 1.62 | 17.1 $\pm$ 0.650 | **16.1 $\pm$ 0.156** |
| | TDI | **7.58 $\pm$ 0.192** | 38.9 $\pm$ 8.76 | **7.21 $\pm$ 0.886** | 8.27 $\pm$ 1.03) | 27.2 $\pm$ 11.1 | **6.59 $\pm$ 0.786** |
| Traffic | MSE | **0.620 $\pm$ 0.010** | 2.52 $\pm$ 0.230 | 1.93 $\pm$ 0.080 | **0.890 $\pm$ 0.11** | 2.22 $\pm$ 0.26 | **1.00 $\pm$ 0.260** |
| | DTW | 24.6 $\pm$ 0.180 | **23.4 $\pm$ 5.40** | **23.1 $\pm$ 0.41** | 24.6 $\pm$ 1.85 | **22.6 $\pm$ 1.34** | 23.0 $\pm$ 1.62 |
| | TDI | **16.8 $\pm$ 0.799** | 27.4 $\pm$ 5.01 | **16.7 $\pm$ 0.508** | 15.4 $\pm$ 2.25 | 22.3 $\pm$ 3.66 | **14.4$\pm$ 1.58** |

Table 1: Forecasting results evaluated with MSE ($\times$100), DTW ($\times$100) and TDI ($\times$10) metrics, averaged over 10 runs (mean $\pm$ standard deviation). For each experiment, best method(s) (Student t-test) in bold.

**MSE comparison:** DILATE outperforms MSE when evaluated on shape (DTW) in all experiments, with significant differences on 5/6 experiments. When evaluated on time (TDI), DILATE also performs better in all experiments (significant differences on 3/6 tests). Finally, DILATE is equivalent to MSE when evaluated on MSE on 3/6 experiments.

**DTW$_\gamma$ [13, 37] comparison:** When evaluated on shape (DTW), DILATE performs similarly to DTW$_\gamma$ (2 significant improvements, 1 significant drop and 3 equivalent performances). For time (TDI) and MSE evaluations, DILATE is significantly better than DTW$_\gamma$ in all experiments, as expected.

We display a few qualitative examples for Synthetic, ECG5000 and Traffic datasets on Fig 4 (other examples are provided in supplementary 2). We see that MSE training leads to predictions that are non-sharp, making them inadequate in presence of drops or sharp spikes. DTW$_\gamma$ leads to very sharp predictions in shape, but with a possibly large temporal misalignment. In contrast, our DILATE predicts series that have both a correct shape and precise temporal localization.

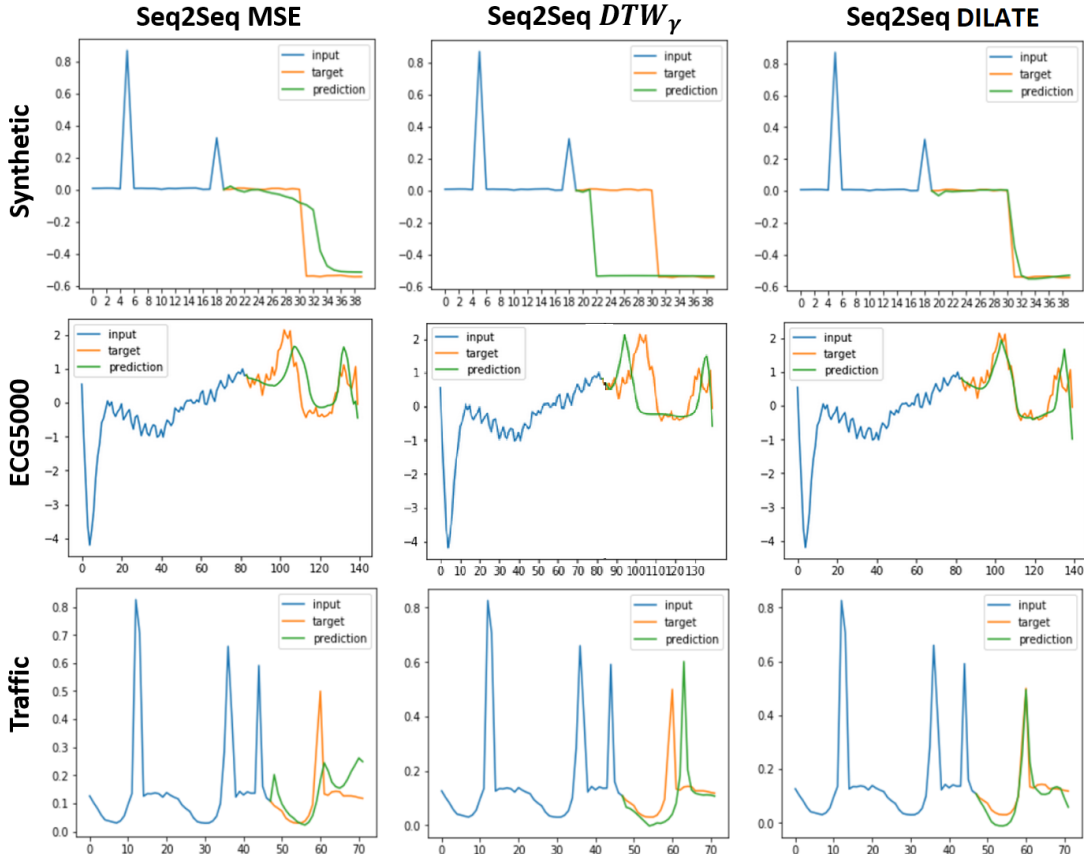

Figure 4: Qualitative forecasting results.

**Evaluation with external metrics** To consolidate the good behaviour of our loss function seen in Table 1, we extend the comparison using two additional (non differentiable) metrics for assessing shape and time. For shape, we compute the ramp score [52]. For time, we perform change point detection on both series and compute the Hausdorff measure between the sets of detected change points $\mathcal{T}^*$ (in the target signal) and $\hat{\mathcal{T}}$ (in the predicted signal):

$$\text{Hausdorff}(\mathcal{T}^*, \hat{\mathcal{T}}) := \max(\max_{\hat{t} \in \hat{\mathcal{T}}} \min_{t^* \in \mathcal{T}^*} |\hat{t} - t^*|, \max_{t^* \in \mathcal{T}^*} \min_{\hat{t} \in \hat{\mathcal{T}}} |\hat{t} - t^*|) \qquad (6)$$

We provide more details about these external metrics in supplementary 1.1.
In Table 2, we report the comparison between Seq2Seq models trained with DILATE, $DTW_\gamma$ and MSE. We see that DILATE is always better than MSE in shape (Ramp score) and equivalent to $DTW_\gamma$ in 2/3 experiments. In time (Hausdorff metric), DILATE is always better or equivalent compared to MSE (and always better than $DTW_\gamma$, as expected).

|  |  | MSE | $DTW_\gamma$ [13] | DILATE (ours) |
|---|---|---|---|---|
|  | Hausdorff | $2.87 \pm 0.127$ | $3.45 \pm 0.318$ | $\mathbf{2.70 \pm 0.166}$ |
| Synthetic | Ramp score ($\times 10$) | $5.80 \pm 0.104$ | $\mathbf{4.27 \pm 0.800}$ | $4.99 \pm 0.460$ |
|  | Hausdorff | $\mathbf{4.32 \pm 0.505}$ | $6.16 \pm 0.854$ | $\mathbf{4.23 \pm 0.414}$ |
| ECG5000 | Ramp score | $\mathbf{4.84 \pm 0.240}$ | $\mathbf{4.79 \pm 0.365}$ | $\mathbf{4.80 \pm 0.249}$ |
|  | Hausdorff | $\mathbf{2.16 \pm 0.378}$ | $\mathbf{2.29 \pm 0.329}$ | $\mathbf{2.13 \pm 0.514}$ |
| Traffic | Ramp score ($\times 10$) | $6.29 \pm 0.319$ | $\mathbf{5.78 \pm 0.404}$ | $\mathbf{5.93 \pm 0.235}$ |

Table 2: Forecasting results of Seq2Seq evaluated with Hausdorff and Ramp Score, averaged over 10 runs (mean $\pm$ standard deviation). For each experiment, best method(s) (Student t-test) in bold.

### 4.3 Comparison to temporally constrained versions of DTW

In Table 3, we compare the Seq2Seq DILATE to its tangled variants Weighted DTW (DILATE$^t$-W) [28] and Band Constraint (DILATE$^t$-BC) [43] on the Synthetic dataset. We observe that DILATE performances are similar in shape for both the DTW and ramp metrics and better in time than both variants. This shows that our DILATE leads a finer disentanglement of shape and time components. Results for ECG5000 and Traffic are consistent and given in supplementary 3. We also analyze the gradient of DILATE *vs* DILATE$^t$-W in supplementary 3, showing that DILATE$^t$-W gradients are smaller at low temporal shifts, certainly explaining the superiority of our approach when evaluated with temporal metrics. Qualitative predictions are also provided in supplementary 3.

| Eval loss |  | DILATE (ours) | DILATE$^t$-W [28] | DILATE$^t$-BC [43] |
|---|---|---|---|---|
| Euclidian | MSE ($\times 100$) | $\mathbf{1.21 \pm 0.130}$ | $1.36 \pm 0.107$ | $1.83 \pm 0.163$ |
| Shape | DTW ($\times 100$) | $\mathbf{23.1 \pm 2.44}$ | $\mathbf{20.5 \pm 2.49}$ | $\mathbf{21.6 \pm 1.74}$ |
|  | Ramp | $\mathbf{4.99 \pm 0.460}$ | $\mathbf{5.56 \pm 0.87}$ | $\mathbf{5.23 \pm 0.439}$ |
| Time | TDI ($\times 10$) | $\mathbf{14.8 \pm 1.29}$ | $17.8 \pm 1.72$ | $19.6 \pm 1.72$ |
|  | Hausdorff | $\mathbf{2.70 \pm 0.166}$ | $\mathbf{2.85 \pm 0.210}$ | $3.30 \pm 0.273$ |

Table 3: Comparison to the tangled variants of DILATE for the Seq2Seq model on the Synthetic dataset, averaged over 10 runs (mean $\pm$ standard deviation).

### 4.4 DILATE Analysis

**Custom backward implementation speedup**: We compare in Fig 5(a) the computational time between the standard Pytorch auto-differentiation mechanism and our custom backward pass implementation (section 3.2). We plot the speedup of our implementation with respect to the prediction length $k$ (averaged over 10 random target/prediction tuples). We notice the increasing speedup with respect to $k$: speedup of $\times 20$ for 20 steps ahead and up to $\times 35$ for 100 steps ahead predictions.

**Impact of** $\alpha$ **(Fig 5(b)):** When $\alpha = 1$, $\mathcal{L}_{DILATE}$ reduces to $DTW_\gamma$, with a good shape but large temporal error. When $\alpha \longrightarrow 0$, we only minimize $\mathcal{L}_{temporal}$ without any shape constraint. Both MSE and shape errors explode in this case, illustrating the fact that $\mathcal{L}_{temporal}$ is only meaningful in conjunction with $\mathcal{L}_{shape}$.

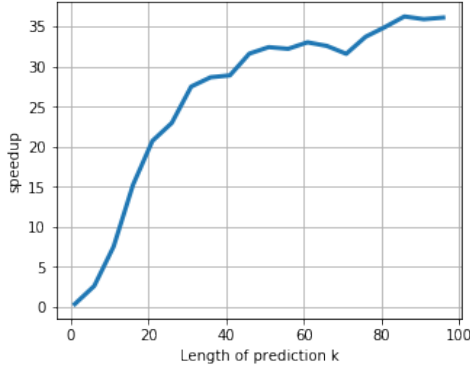

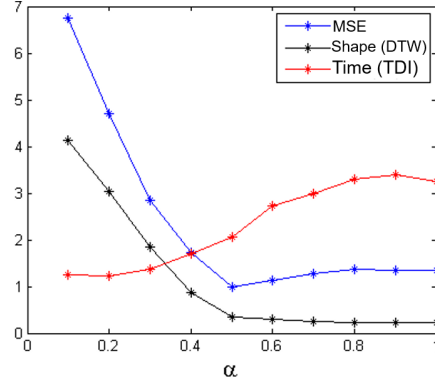

| Figure 5(a): Speedup of DILATE | Figure 5(b): Influence of $\alpha$ |
|---|---|

## 4.5 Comparison to state of the art time series forecasting models

Finally, we compare our Seq2Seq model trained with DILATE with two recent state-of-the-art deep architectures for time series forecasting trained with MSE: LSTNet [30] trained for one-step prediction that we apply recursively for multi-step (LSTNet-rec) ; and Tensor-Train RNN (TT-RNN) [60] trained for multi-step[2]. Results in Table 4 for the traffic dataset reveal the superiority of TT-RNN over LSTNet-rec, which shows that dedicated multi-step prediction approaches are better suited for this task. More importantly, we can observe that our Seq2Seq DILATE outperforms TT-RNN in all shape and time metrics, although it is inferior on MSE. This highlights the relevance of our DILATE loss function, which enables to reach better performances with simpler architectures.

| Eval loss | | LSTNet-rec [30] | TT-RNN [60, 61] | Seq2Seq DILATE |
|---|---|---|---|---|
| Euclidian | MSE (x100) | $1.74 \pm 0.11$ | $\mathbf{0.837 \pm 0.106}$ | $1.00 \pm 0.260$ |
| Shape | DTW (x100) | $42.0 \pm 2.2$ | $25.9 \pm 1.99$ | $\mathbf{23.0 \pm 1.62}$ |
| | Ramp (x10) | $9.00 \pm 0.577$ | $6.71 \pm 0.546$ | $\mathbf{5.93 \pm 0.235}$ |
| Time | TDI (x10) | $25.7 \pm 4.75$ | $17.8 \pm 1.73$ | $\mathbf{14.4 \pm 1.58}$ |
| | Hausdorff | $\mathbf{2.34 \pm 1.41}$ | $\mathbf{2.19 \pm 0.125}$ | $\mathbf{2.13 \pm 0.514}$ |

Table 4: Comparison with state-of-the-art forecasting architectures trained with MSE on Traffic, averaged over 10 runs (mean $\pm$ standard deviation).

## 5 Conclusion and future work

In this paper, we have introduced DILATE, a new differentiable loss function for training deep multi-step time series forecasting models. DILATE combines two terms for precise shape and temporal localization of non-stationary signals with sudden changes. We showed that DILATE is comparable to the standard MSE loss when evaluated on MSE, and far better when evaluated on several shape and timing metrics. DILATE compares favourably on shape and timing to state-of-the-art forecasting algorithms trained with the MSE.

For future work we intend to explore the extension of these ideas to probabilistic forecasting, for example by using bayesian deep learning [21] to compute the predictive distribution of trajectories, or by embedding the DILATE loss into a deep state space model architecture suited for probabilistic forecasting. Another interesting direction is to adapt our training scheme to relaxed supervision contexts, *e.g.* semi-supervised [42] or weakly supervised [16], in order to perform full trajectory forecasting using only categorical labels at training time (*e.g.* presence or absence of change points).

**Aknowledgements**   We would like to thank Stéphanie Dubost, Bruno Charbonnier, Christophe Chaussin, Loïc Vallance, Lorenzo Audibert, Nicolas Paul and our anonymous reviewers for their useful feedback and discussions.

## Footnotes

[1]A Seq2Seq architecture was the winner of a 2017 Kaggle competition on multi-step time series forecasting (https://www.kaggle.com/c/web-traffic-time-series-forecasting)

[2]We use the available Github code for both methods.

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
