[Supplementary Material]

# Shape and Time Distortion Loss for Training Deep Time Series Forecasting Models Supplementary Material

**Vincent Le Guen** [1,2]
vincent.le-guen@edf.fr

**Nicolas Thome** [2]
nicolas.thome@cnam.fr

(1) EDF R&D
6 quai Watier, 78401 Chatou, France

(2) CEDRIC, Conservatoire National des Arts et Métiers
292 rue Saint-Martin, 75003 Paris, France

## 1 Experimental setup

### 1.1 Metrics

We describe here the two external metrics used in our experiments to evaluate the shape and timing errors.

**Ramp score**  The notion of *ramping event* is a major issue for intermittent renewable energy production that needs to be anticipated for electricity grid management. For assessing the performance of trained forecasting models in presence of ramps, the Ramp Score was proposed in (VCP[+]17). This score is based on a piecewise linear approximation on both input and target time series by the Swinging Door algorithm (Bri90; FHO13). The Ramp Score described in (VCP[+]17) is computed as the integral between the unsigned difference of derivatives of both linear approximated series. For assessing only the shape error component, we apply in our experiments the ramp score on the target and prediction series after alignment by the optimal DTW path.

**Hausdorff distance**  Given a set of change points $\mathcal{T}^*$ in the target signal and change points $\hat{\mathcal{T}}$ in the predicted signal, the Hausdorff distance is defined as:

$$\text{Hausdorff}(\mathcal{T}^*, \hat{\mathcal{T}}) := \max(\max_{\hat{t} \in \hat{\mathcal{T}}} \min_{t^* \in \mathcal{T}^*} |\hat{t} - t^*|, \max_{t^* \in \mathcal{T}^*} \min_{\hat{t} \in \hat{\mathcal{T}}} |\hat{t} - t^*|) \tag{1}$$

It corresponds to the greatest temporal distance between a change point and its prediction.

We now explain how the change points are computed for each dataset: for Synthetic, we know exactly by construction the positions of the change points in the target signals. For the predictions, we look for a single change point corresponding to the location of the predicted step function. We use the exact segmentation method by dynamic programming described in (TOV18) with the Python toolbox http://ctruong.perso.math.cnrs.fr/ruptures-docs/build/html/index.html# .

For ECG5000 and Traffic datasets which present sharp peaks, this change point detection algorithm is not suited (detected change points are often located at the inflexion points of peaks and not at the exact peak location). We thus use a simple peak detection algorithm based on first order finite differences. We tune the threshold parameter for outputting a detection and the min distance between detections parameter experimentally for each dataset.

## 2 Experiments

We provide here additional qualitative predictions for all datasets.

Figure 1: Qualitative predictions for the Synthetic dataset.

Figure 2: Qualitative predictions for the ECG5000 dataset.

Figure 3: Qualitative predictions for the Traffic dataset.

## 3 Comparison to the temporally constrained variants of DTW

To understand the difference of behaviour between DILATE and its variant DILATE$^t$-W, we build a synthetic example by starting with a target step function as in Fig 4 (b) and a perfect step prediction. Then we translate progressively the step prediction and we compute three error metrics (Fig 4 (a)). As expected, the DTW error remains zero since we only have temporal shift. DILATE and DILATE$^t$-W increasingly penalize the temporal shift but the DILATE$^t$-W curve is flatter around 0 (lower gradient for the small temporal shifts). This suggests that DILATE is better suited to recover finer temporal

alignment, as the quantitative results in Table 1. We provide as illustration two experimental results on Fig 4 (b) and (c) on the Synthetic dataset: DILATE leads to better temporal localization.

(a)  (b) DILATE$^t$-W  (c) DILATE

Figure 4: Qualitative comparison between DILATE and DILATE$^t$-W.

We report in Table 1 the full quantitative results on the 3 datasets:

| Dataset | Eval loss | | DILATE (ours) | DILATE$^t$-W (JJO11) | DILATE$^t$ BC (SC90) |
|---|---|---|---|---|---|
| Synth | Euclidian | MSE ($\times 100$) | **1.21 $\pm$ 0.130** | 1.36 $\pm$ 0.107 | 1.83 $\pm$ 0.163 |
| | Shape | DTW ($\times 100$) | 23.1 $\pm$ 2.44 | **20.5 $\pm$ 2.49** | **21.6 $\pm$ 1.74** |
| | | Ramp | **4.99 $\pm$ 0.460** | 5.56 $\pm$ 0.87 | 5.23 $\pm$ 0.439 |
| | Time | TDI (($\times 10$) | **14.8 $\pm$ 1.29** | 17.8 $\pm$ 1.72 | 19.6 $\pm$ 1.72 |
| | | Hausdorff | **2.70 $\pm$ 0.16** | **2.85 $\pm$ 0.210** | 3.30 $\pm$ 0.273 |
| ECG | Euclidian | MSE ($\times 100$) | **30.3 $\pm$ 4.10** | **31.1 $\pm$ 4.51** | 66.2 $\pm$ 11.4 |
| | Shape | DTW ($\times 10$) | **16.1 $\pm$ 0.156** | 17.9 $\pm$ 2.09 | 17.6 $\pm$ 0.20 |
| | | Ramp | **4.80 $\pm$ 0.249** | **4.63 $\pm$ 0.307** | **4.49 $\pm$ 0.255** |
| | Time | TDI ($\times 10$) | **6.59 $\pm$ 0.786** | 8.08 $\pm$ 0.845 | 16.8 $\pm$ 3.09 |
| | | Hausdorff | 4.23 $\pm$ 0.414 | **3.56 $\pm$ 0.305** | 6.81 $\pm$ 1.41 |
| Traffic | Euclidian | MSE ($\times 100$) | **1.00 $\pm$ 0.260** | **1.06 $\pm$ 0.12** | 1.87 $\pm$ 0.25 |
| | Shape | DTW ($\times 100$) | **23.0 $\pm$ 1.62** | **22.2 $\pm$ 1.07** | **22.6 $\pm$ 2.01** |
| | | Ramp | 5.93 $\pm$ 0.235 | 5.92 $\pm$ 0.177 | **5.63 $\pm$ 0.46** |
| | Time | TDI (($\times 10$) | **14.4 $\pm$ 1.58** | **15.5 $\pm$ 1.50** | 17.6 $\pm$ 0.23 |
| | | Hausdorff | **2.13 $\pm$ 0.514** | **1.79 $\pm$ 0.437** | **2.08 $\pm$ 0.331** |

Table 1: Comparison to the tangled variants of DILATE. Results averaged over 10 runs (mean $\pm$ standard deviation). For each experiment, best method(s) (Student t-test) in bold.