[Reviews · NeurIPS 2019]

Reviewer 1



This paper proposes an interesting idea in order to efficiently forecast non-stationary time series at multiple times ahead. In order to achieve this, the authors introduce an objective function called Shape and Time Distorsion Loss (STDL) to train deep neural network. The paper is well written, clear, and of certain significance. In Figure 1 the authors illustrate the limitation of certain existing approaches as a way to motivate their contributions. I am not convinced with the arguments presented there. 1) The use of online regression with adaptive tuning parameters can accurately predict the target provided in Figure 1 (e.g. Recursive Least Squares with Forgetting Factors, Kalman Filter with adaptive noise and so on...) without having recourse to new approaches. 2) Those algorithms are missing in the related art section. Furthermore, the authors mention that "Experiments carried out on several synthetic and real non-stationary datasets reveal that models trained with STDL significantly outperform models trained with the MSE loss function when evaluated with shape and temporal distorsion metrics, while SDTL maintains very good performance when evaluatedwith MSE.", this is a strong statement as the authors do not compare their approach with online approaches capable of dealing with non-stationary time series. However, the approach to training deep neural networks with the shape and time distorsion loss is interesting but deserves to be compared with competing approaches that showed promised for non-stationary forecasting.

Reviewer 2



1. The distortion problem is well-known on the time-series data. This paper aims to propose a new objective function named Shape and Time Distortion Loss (STDL) to address the problem of time series forecasting for non-stationary signals and multiple future steps prediction. The proposed loss is useful for researchers to train an end-to-end forecasting model that can handle the shape and time distortion error in a convenient way. 3. The results of this work show the proposed loss significantly outperforms other baseline loss functions such as MSE and DTW. Moreover, the proposed loss function is tested in the end of this paper to point that the loss function helps smaller models to perform as well as some other larger models. 4. Moreover, this work points the complexity of the computation of this loss by their custom backward pass is in O(nm). The implementation time between standard Pytorch auto-differentiation mechanism and the custom backward pass implementation has a significant speedup with respect to the prediction length k.

Reviewer 3



It is hard to say what constitutes a good prediction, and loss functions matter more than model architecture development now that we have neural networks and auto-differentiation. All the losses we use are some proxies inspired by loss functions in ML that were developed because they were easy to optimize a model for. So adding new, intuitively appealing loss function is significant. The paper is well written and enjoyable to read. The evaluation is fairly comprehensive and looks soundly executed. For completeness, you might use STDL as an evaluation loss as well. Are you able to summarize generic conclusions about features of the time series that will be the individual losses train well vs poorly? E.g. "MSE rounds sharp level changes". It puzzles me why people consider ECG a time series prediction task. It is commonly used as a dataset but in what real world scenario would you want to predict it? How fast is the computation? The dynamic programming looks pretty complex. You only give the speedup - what is the absolute number? Is its evaluation of the loss a bottleneck in the learning? How do you set \delta s in your experiments?

[Author Response · NeurIPS 2019]

We thank the reviewers for their meaningful and valuable comments, which help to improve the quality of our work.

**R1 on paper motivation:** Figure 1 in our submission illustrates the limitation of using MSE as a loss for training deep
forecasting models. Since MSE is similar for the three predictions in (a), (b) and (c), a gradient-based optimization is
unable to produce training signals preferring predictions (b) and (c) over (a). By no means we wanted to claim that
certain existing approaches are unable to perform step prediction in this simple example. This work does not focus
on designing new forecasting models, but introduces the STDL loss function as an alternative to MSE. STDL is thus
model-agnostic and can be used for training various forecasting models - as shown in experiments and below.
**R1 on state-of-the-art methods:** We thank R1 for suggesting to compare our results to online regression with adaptive
parameters (forgetting recursive least squares, adaptive Kalman filters). Although these are historical approaches for
Bayesian inference in State Space Models (SSMs), their direct application to multi-step forecasting is not straightforward
because they require input data for adaptation at each time step. For this reason, several state-of-the art multi-step
approaches combine SSMs and deep learning based on Seq2Seq architectures [1, 2, 3, 4].
To fulfill R1 requests, we perform additional experiments (shown in blue) on the Traffic dataset (Table 4 in submission).
The results of the Deep AR baseline[1] (obtained with GitHub code) is still outperformed by a simple Seq2Seq model
trained with STDL (results shown in submission, column 4 in black), and equivalent in temporal metrics. Training Deep
AR with STDL would be an interesting future exploration. Finally, we provide results when training the recent TT-RNN
(refs [48,49] in submission) with STDL, reaching the best shape and temporal performances. These new results further
highlight the importance of STDL ; we will be glad to add these comparisons in the final paper if accepted.

| Eval loss | | LSTNet-rec (MSE) | TT-RNN (MSE) | Deep AR (MSE) | Seq2Seq (STDL) | TT-RNN (STDL) |
|---|---|---|---|---|---|---|
| Euclidian | MSE (x100) | $1.74 \pm 0.11$ | $\mathbf{0.840 \pm 0.106}$ | $0.966 \pm 0.068$ | $1.00 \pm 0.260$ | $\mathbf{0.930 \pm 0.09}$ |
| Shape | DTW (x100) | $42.0 \pm 2.2$ | $25.9 \pm 1.99$ | $27.8 \pm 1.55$ | $23.0 \pm 1.62$ | $\mathbf{21.4 \pm 0.79}$ |
| | Ramp (x10) | $9.00 \pm 0.577$ | $6.71 \pm 0.546$ | $7.56 \pm 0.42$ | $5.93 \pm 0.235$ | $\mathbf{5.27 \pm 0.27}$ |
| Time | TDI (x10) | $25.7 \pm 4.75$ | $17.8 \pm 1.73$ | $\mathbf{14.6 \pm 0.94}$ | $\mathbf{14.4 \pm 1.58}$ | $15.7 \pm 1.02$ |
| | Hausdorff | $2.34 \pm 1.41$ | $2.19 \pm 0.12$ | $2.04 \pm 0.11$ | $2.13 \pm 0.514$ | $\mathbf{1.88 \pm 0.153}$ |

**R2 on more complex datasets:** As requested, we provide additional experiments on 2 more complex datasets:
household electricity consumption and solar energy. The former corresponds to a multivariate forecasting problem
involving 10 exogenous input variables (global intensity, voltage, sub-metering, date, *etc*), requiring the extraction of
complex interactions in data for spiky patterns prediction. The latter has very fine time granularity (10min vs 1h for
Traffic), needing to extract accurate time features. The results shown below again illustrate the superiority of training
Seq2Seq models with SDTL compared to MSE.

| Method | Household electricity consumption | | | Solar energy | | |
|---|---|---|---|---|---|---|
| | MSE (x10) | DTW | TDI | MSE (x1000) | DTW (x100) | TDI (x10) |
| Seq2Seq MSE | $\mathbf{18.3 \pm 2.5}$ | $4.54 \pm 0.40$ | $\mathbf{2.49 \pm 0.26}$ | $\mathbf{13.7 \pm 1.5}$ | $24.3 \pm 3.4$ | $12.9 \pm 1.4$ |
| Seq2Seq STDL | $\mathbf{19.9 \pm 2.4}$ | $\mathbf{3.85 \pm 0.26}$ | $\mathbf{2.30 \pm 0.59}$ | $\mathbf{14.4 \pm 0.57}$ | $\mathbf{20.9 \pm 1.1}$ | $\mathbf{5.71 \pm 0.83}$ |

**R2 on $\alpha$ tuning:** $\alpha$ is chosen on a validation set, by selecting the lowest value for which $\mathcal{L}_{shape}$ gets comparable
performance than a reference $DTW_\gamma$ trained model. This setup will be added in the final version if accepted.

**R3 on reporting STDL as evaluation metric:** these results will be added in our tables if accepted.
**R3 on training time:** 1 training epoch with our Seq2Seq GRU network takes about 0.5s for MSE vs 1.7s for SDTL on
Synthetic (1s vs 8s on ECG5000, 3s vs 33s on Traffic). The overhead is due to the sequential computation of the STDL
(dynamic programming in forward and backward passes). Note that no overhead is involved at test time.
**R3 on choice of $\delta$:** We choose $\delta$ as the euclidean loss (paper l. 102-103), which is common for computing DTW, but
any other distance (*e.g.* mean absolute error) could be employed.
**R3 on code sharing:** source code will be made available on GitHub after acceptance.
**R3 on ECG:** predicting the shape and time interval between heartbeats could be helpful for cardiologists, to detect
abnormal heartbeats such as 'premature ventricular contraction'.
**R3 on feature interpretation:** understanding the effects of our shape and time loss terms on the learned features is an
interesting but non trivial perspective. A possible direction to this end is to use feature visualization techniques [5].
**R3 on confidence intervals:** we could use MC Dropout (Gal *et. al.*, ICML'16) to compute the predictive distribution
of trajectories ; or embed the STDL loss in a deep SSM architecture suited for probabilistic forecasting.

[1] D. Salinas, V. Flunkert, J. Gasthaus. "DeepAR: Probabilistic forecasting with autoregressive recurrent networks", ICML 2017
[2] S. Rangapuram, M. Seeger, J. Gasthaus, L. Stella, Y. Wang, "Deep state space models for time series forecasting", NeurIPS2018
[3] X.Jin, S.Li, Y. Zhang, X.Yan, "Multi-step deep autoregressive forecasting with latent states", ICML 2019 Time Series Workshop
[4] Y. Wang, A. Smola, D. C Maddix, J. Gasthaus, D. Foster,T. Januschowski. "Deep factors for forecasting", ICML 2019
[5] Andrej Karpathy, Justin Johnson, and Li Fei-Fei. "Visualizing and understanding recurrent networks", ICLR 2016


[Meta-Review · NeurIPS 2019]

The paper presents an approach to time-series forecasting, which the reviewers thought is interesting and sound. Some of the concerns on the experimental side raised by the reviewers were successfully addressed during the rebuttal.